# Rare pathogenic structural variants show potential to enhance prostate cancer germline testing for African men

Tingting Gong [1,2], Jue Jiang [1], Korawich Uthayopas[1], M. S. Riana Bornman [3], Kazzem Gheybi[1], Phillip D. Stricker [4], Joachim Weischenfeldt [5,6], Shingai B. A. Mutambirwa[7], Weerachai Jaratlerdsiri[1] & Vanessa M. Hayes [1,3,8] ✉

Prostate cancer (PCa) is highly heritable, with men of African ancestry at greatest risk and associated lethality. Lack of representation in genomic data means germline testing guidelines exclude for Africans. Established that structural variations (SVs) are major contributors to human disease and prostate tumourigenesis, their role is under-appreciated in familial and therapeutic testing. Utilising clinico-methodologically matched deep-sequenced whole-genome data for 113 African *versus* 57 European PCa patients, we interrogate 42,966 high-quality germline SVs using a best-fit pathogenicity prediction workflow. We identify 15 potentially pathogenic SVs representing 12.4% African and 7.0% European patients, of which 72% and 86% met germline testing standard-of-care recommendations, respectively. Notable African-specific loss-of-function gene candidates include DNA damage repair *MLH1* and *BARD1* and tumour suppressors *FOXP1, WASF1* and *RB1*. Representing only a fraction of the vast African diaspora, this study raises considerations with respect to the contribution of kilo-to-mega-base rare variants to PCa pathogenicity and African-associated disparity.

Prostate cancer (PCa) is a significant global health burden and a leading cause of male-associated cancer deaths[1]. With one of the highest heritability rates (estimated 58%), PCa risk shows a great degree of variability[2], particularly when considering a man's ancestral heritage. In the United States, Black men are at greatest risk for aggressive disease presentation[3] and, depending on age at diagnosis over double to triple (< 65 years) the risk for PCa-associated mortality than White Americans[4,5]. Contributed by a complex interaction of socioeconomic factors and genetics[6], inherited risk includes a combination of both common (low-risk with combined genetic risk scores) and rare (high-risk or pathogenic) germline variants[7,8]. Revolutionised through the advancement of precision oncology, most notably the approval of the poly-(ADP ribose) polymerase (PARP) inhibitors Olaparib[9] and Rucaparib[10] for the treatment of metastatic castrate-resistant PCa for patients harbouring rare pathogenic variants in specified DNA repair genes[11], has increased the value for germline testing. Furthermore, the National Comprehensive Cancer Network (NCCN) recommends germline testing for all men with metastatic, recurrent or high-risk localised PCa, regardless of family history[12]. Although a significant risk factor for aggressive disease, no consensus could be reached for men of African ancestry[13], while a recent review further highlighted the knowledge gap[14].

[1]Ancestry and Health Genomics Laboratory, Charles Perkins Centre, School of Medical Sciences, Faculty of Medicine and Health, University of Sydney, Camperdown, NSW 2050, Australia. [2]Human Phenome Institute, Fudan University, Shanghai, China. [3]School of Health Systems and Public Health, University of Pretoria, Pretoria, South Africa. [4]St Vincent's Prostate Cancer Research Centre, Sydney, NSW, Australia. [5]Finsen Laboratory, Rigshospitalet, DK-2200 Copenhagen, Denmark. [6]Biotech Research & Innovation Centre, University of Copenhagen, DK-2200 Copenhagen, Denmark. [7]Department of Urology, Sefako Makgatho Health Science University, Dr George Mukhari Academic Hospital, Medunsa, Ga-Rankuwa, South Africa. [8]Manchester Cancer Research Centre, University of Manchester, Manchester M20 4GJ, UK. ✉e-mail: vanessa.hayes@sydney.edu.au

The lack of consensus for PCa germline testing in Black men is directly attributed to a lack of available data, compounded by a lack of African-relevant genomic data that captures the true extent of elevated genetic diversity. While consensus has yet to be reached for minority inclusion in the benefits of recent breakthroughs in PCa precision oncology, contradictory studies suggest that Black American patients harbour more[15] and conversely less actionable pathogenic variants than White Americans[16]. The picture is no different for Africa, although more recently PCa genomics has reached the continent, with the inclusion of whole exome (*n* = 45 Nigerian)[17] and whole genome sequencing studies (*n* = 113 Black South Africans)[18]. Although preliminary, notable differences within Africa are emerging. For example, an elevated frequency of *BRCA1* germline mutations reported for Nigerian patients, reflecting African American data[17,19], is lacking in Southern African cases[20]. In addition, we have recently reflected on the lack of the West African exclusive and functionally relevant common PCa susceptibility variants *CHEK2* p.Ile448Ser (rs17886163) and *HOXB13* p.Ter285Lys (rs77179853) in Southern Africa[21,22]. Reporting a 2.1-fold age-adjusted increase in aggressive PCa presentation in Black South African *versus* Black American men[23], through deep sequenced interrogation for the 20 most common genes included in PCa germline testing panels using NCCN inclusion criteria (Gleason score ≥ 8), we observed a prevalence for rare pathogenic variants of 5.6%[20], comparable with a single East African study (5.7%)[24] and almost half that reported for non-Africans (11.8%)[25]. These studies highlight the need for developing African-relevant PCa germline testing panels through African inclusion in genome profiling.

Again, it is well established that Structural Variations (SVs) play a critical role in prostate tumour progression with prognostic and therapeutic potential[26,27], including tumours derived from men of African ancestry[18,28]. Yet, irrespective of patient ancestry, little is known with regard to the contribution of germline potentially pathogenic rare SVs. Typically, greater than 50 bases in length, SVs encompassing large deletions (DEL), duplications (DUP), insertions (INS), inversions (INV) and translocations (TRA), are overlooked and/or difficult to resolve using current germline genetic testing assays. While it is well established that SVs play a critical role in diagnostic screening for inherited genetic diseases[29], more recently, long-read sequencing has been used to identify potential pathogenic SVs in hereditary cancer syndromes[30] and known breast cancer susceptibility genes[31], however, the impact of rare pathogenic SVs on PCa predisposition, and in turn targeted treatment, remains unknown.

Here, expanding on our earlier work[18,20,28], including deep sequenced germline genomes for 113 African (Black South African) and 57 European (4 South African, 53 Australian) PCa patients, through high-quality SV calling and genotyping, comprehensive gene annotation and best-fit pathogenicity prediction, we interrogate for rare potentially pathogenic SVs (PP-SVs). While agreeably a small study size, this resource is not only unique for the African continent, importantly it provides clinically and technically matched non-African data for direct comparative analyses, while our whole genome approach increases sensitivity for SV detection. We provide computational and case-associated expression evidence for PP-SV contribution to aggressive PCa presentation and associated ancestral disparities, including unknown to PCa, ancestry-specific germline testing gene candidates. Our study reveals the added value for whole-genome germline SV interrogation and African inclusion to provide important insights into optimising PCa germline testing for global impact.

## Results
### NCCN high-risk characterisation for ancestrally assigned PCa patients
Clinically and technically matched whole genome sequenced germline data (mean coverage 45.9X; range 30.2–97.6X) was derived from whole blood from 170 PCa patients, ancestrally classified previously using 7,472,833 genome-wide SNVs and population substructure analysis[18]. In brief, 113 Black South African patients presented with an African ancestral genetic fraction of > 85%, while the 57 White patients presented with European ancestral genetic fractions of > 90% (4 South African, 52 Australian) and 73.7% European and 26.3% Asian sub-structure (1 Australian) (Supplementary Table 1). Importantly, although the mean age was 5 years younger at presentation or surgery, a greater number of European (86%; 49/57) over African patients (72%; 81/113) met current NCCN guidelines for germline testing based on International Society of Urological Pathology (ISUP) Group Grading defined as high-risk localised PCa (ISUP 4/5 or Gleason score ≥ 8). Notably, we have previously provided evidence for the extension of these criteria for Black South African men to include ISUP 3, which would expand our cohort of high-risk Black men to 82% (93/113)[20]. While Black South Africans present with significantly elevated median and range of prostate-specific antigen (PSA) levels (median 244 ng/mL *versus* 9.4), as previously presented[18,23], still the study was biased towards over-representation of NCCN guidelines for PSA-inclusive high-risk PCa for the European (70.2%; 40/57) over African patients (65/113; 57.5%).

### Genome-wide gene-disrupting SV discovery
In this study, we identified and genotyped 42,966 high-quality germline SVs. We found a median of 9206 SVs (range: 8891– 9708) per African genome, which is significantly higher than the median of 7490 (range: 7309–8050) per European genome (*p*-value = 1.1e-26 by Wilcoxon test). In total, we identified 38,668 African-derived SVs (18,674 private) and 24,292 European-derived SVs (4298 private) (Supplementary Table 2). Including only high-quality genotype calls for allele frequency (AF) estimation left a total of 33,243 high-confidence SVs. Excluding common SVs, defined as minor allele frequency (MAF) > 5%, a total of 20,982 rare (MAF ≤ 1%) and low-frequency (MAF = 1 to 5%) SVs remained across the ancestries for further annotation (Fig. 1).

Further interrogation for gene regions overlapping, we identified 1857 gene-disruptive SVs, including 1752 potential Loss-of-Function (pLoF), 52 Copy Gain (CG) and 53 Intragenic Exon DUP (IED) (detailed in Methods). Notably, pLoF, CG and IED SVs can have a functional impact on genes through either gene inactivation or increased dosage effect[32]. Conversely, there is no clear or direct coding effect by SVs with other gene impact types, which included in our study 109 partial gene DUP, 22 partial exons DUP, 48 whole-gene INV, 343 promoter SVs, 9431 intronic SVs and 258 enhancer SVs. As such, the latter SVs were not discussed further. In total, we identified 1857 (MAF ≤ 5%) gene-disruptive SVs of which 1407 are African-relevant, including 93% (1314) African-private, and 543 European-relevant, including 83% (450) European-private (Supplementary Table 2). There were 93 SVs (5%) shared by both African and European PCa patients. The 1857 gene-disruptive SVs (1050 rare in both African and European) underwent further downstream interrogation for potential clinical relevance. Of the 1857 gene-disruptive SVs, 1167 were previously reported in the dbVar database of SVs, while 690 were absent, of which 513 (74%) are uniquely African (Fig. 1).

### Characterising ClinVar verified candidate potentially pathogenic SVs
Of the 1167 dbVar-reported gene-disruptive SVs, 14 (1.2%) were recorded in ClinVar, with three reported as 'pathogenic' or 'likely pathogenic' based on functional prediction consensus. One 2958 bp likely pathogenic DEL results in loss of exon 7 in *OCA2* (Supplementary Fig. 1), a 5064 bp pathogenic DEL leads to exon 5–7 loss in *PIGN* (Supplementary Fig. 2), while a 235 bp likely pathogenic DUP duplicates exon 3 of *SLC3A1* (Supplementary Fig. 3). The *OCA2* and *PIGN* DELs were identified in a single African patient each, while the *SLC3A1* DUP presented in two African patients (Table 1).

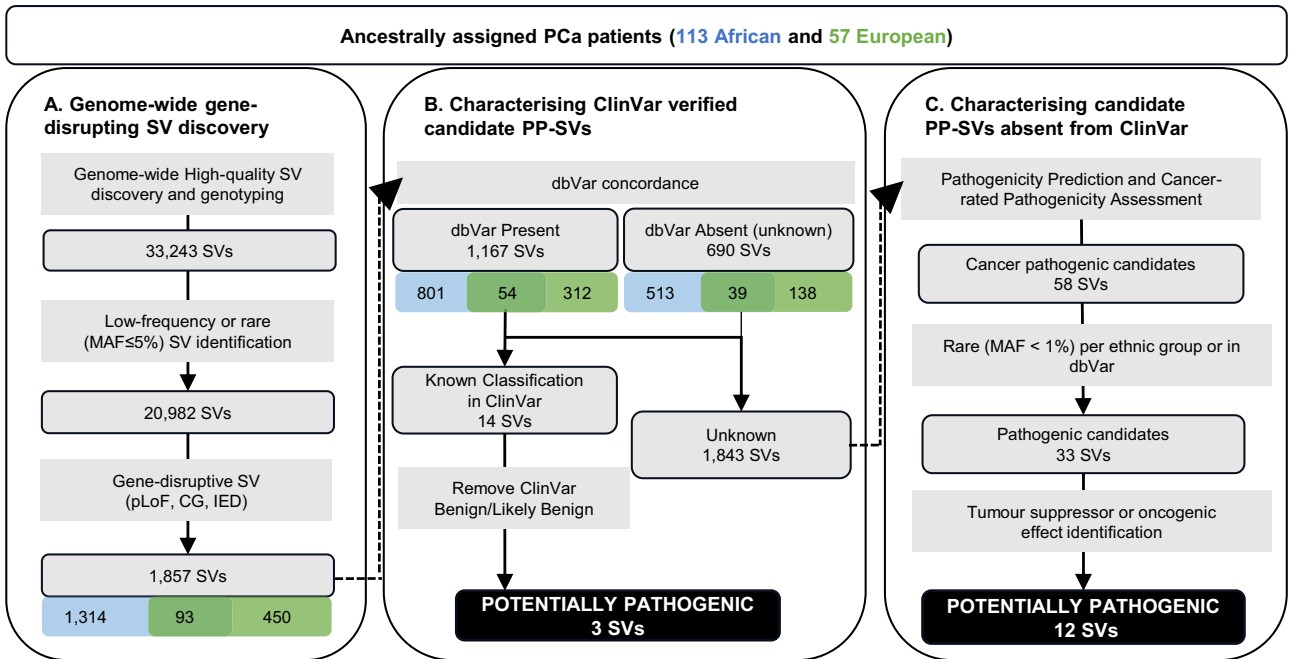

**Fig. 1 | Workflow of PCa potentially pathogenic SV (PP-SV) identification.** Including genome-wide gene-disrupting SV discovery (**A**) identifying known potential pathogenic SVs (**B**), while further characterising SVs of unknown significance (**C**). The detailed criteria to predict the potential pathogenicity were shown in Supplementary Table 4. The identification of tumour suppressor or oncogenic effect for disrupted genes by pathogenic candidates and the related literature were shown in Supplementary Table 7.

Although pathogenic in ClinVar, none have been associated with cancer phenotypes and include rather oculocutaneous albinism, multiple congenital anomalies-hypotonia-seizures syndrome and cystinuria, respectively. As such, we searched the literature for plausibility with further ascertainment derived from normal prostate and tumour tissue data sets using GENT2[33]. Reported to be downregulated in numerous cancer types (all-type *p*-value < 0.001, GENT2 T-test), although not significant for PCa, pLoF deletion of the pigmentation gene *OCA2* has been linked not only to Prader-Willi syndrome, but also Prader-Willi associated malignancies[34], and melanoma[35], with recent studies linking melanoma with increased PCa risk[36]. Highly expressed in normal prostate tissue with significant upregulation in tumour tissue (*p*-value < 0.001, GENT2 T-test), *PIGN* functions as a cancer chromosomal instability suppressor gene[37,38]. Although at lower levels, *SLC3A1* is also upregulated in PCa (*p*-value < 0.001, GENT2 T-test), with overexpression in breast cancer associated with tumourigenesis[39]. These observations, taken together, provide the rational for characterising the pLoF *OCA2* and *PIGN* DELs and *SLC3A1* IED as potentially pathogenic SVs (PP-SVs). Notably, all three SVs are reported as rare (irrespective of ancestry) in multiple population-wide studies including gnomAD SV[32], 1000 genomes Project (1KGP)[40,41] and TOPMed SV[42] (Supplementary Data 1). However, the *OCA2* PP-SV was observed at a low frequency (1% <MAF ≤ 5%) in our Southern African population-matched control data including whole genomes derived from 49 younger aged ( < 45 years) largely female (41, age range: 18–44) over male participants (8, age range: 18–39). Further gene interrogation showed an additional pLoF TRA on *OCA2*, resulting in *DNAH9-OCA2* gene fusion (Supplementary Table 3).

**Characterising candidate potentially pathogenic SVs absent from ClinVar**
Among 1843 SVs with unknown classification in ClinVar or absent from dbVar, we predicted their potential pathogenicity based on four SV impact prediction tools, including StrVCTVRE[43], CADD-SV[44], POSTRE[45] and PhenoSV[46]. The number of scored SVs by four tools and their types were shown in Supplementary Fig. 4 and Supplementary Table 4.

Candidate SVs were required to meet two of the following criteria: StrVCTVRE score ≥ 0.37, CADD-SV score ≥ 10, POSTRE score ≥ 0.8 and/or PhenoSV score ≥ 0.5 (Supplementary Table 5 and Methods). Based on this criterion, all three ClinVar identified pathogenic or likely pathogenic SVs and the single SV of uncertain significance were successfully annotated as pathogenic candidates, while conversely, our workflow excluded for all 10 ClinVar characterised benign SVs (Supplementary Table 6). Using our criteria, 291 SVs were defined as PP-SV candidates (107 DELs, 16 DUPs, 11 INVs and 157 TRAs) disrupting 419 genes. In total, 190 candidate SVs were private to African and 88 to European patients, with 13 shared between the ancestries (Supplementary Table 5).

To further define cancer-related pathogenic potential, we assessed for the presence of disrupted genes by PP-SV candidates in gene sets derived from the Human Molecular Signature Database (MSigDB) oncogenic signature and hallmark gene sets[47] and COSMIC Cancer Gene Census (COSMIC CGC) cancer driver genes[48]. Requiring disrupted genes in two of the three cancer gene sets, 58 SVs were defined as cancer-related PP-SV candidates, including 20 DELs, 3 DUPs, 6 INVs and 29 TRAs, disrupting 56 genes. Of the 58 candidates, 23 of them were identified with MAF between 1% to 5% in either African or European patients, leaving 35 rare PP-SV candidates for further consideration, of which 16 have been reported in dbVar. Two dbVar SVs, including TRA disrupting gene *NBEA* and *POLR2C* DEL, were reported at low frequencies (AF = 0.03 and 0.01, respectively) (Supplementary Data 1) and were therefore excluded from further analysis. Using our criteria, 33 rare cancer-related PP-SV candidates were identified (Supplementary Data 2 and Fig. 1), including 15 DELs, 3 DUPs (1 IED and 2 CGs), 5 INVs and 10 TRAs.

Of the 15 pLoF DELs, 11 were excluded as PP-SVs, with impacting genes showing oncogenic behaviour in multiple cancer types or no strong evidence for their tumour suppressor effects (Supplementary Table 7). Conversely, four pLoF DELs were defined as PP-SVs, impacting known tumour suppressors or established DNA damage repair genes (Supplementary Table 7). Two of them are known to dbVar, including a *SLC7A2* 125,146 bp DEL identified in two African

**Table 1 | Candidate potentially pathogenic (PP) SVs identified in 170 PCa patients**

| Genes | Gene impact type[1] | 1stChr | pos1 | 2ndChr | pos2 | SV type | ClinVar / dbVar concordance | MAF African (this study) | MAF African (control)[2] | MAF European (this study) | MAF African (dbVar)[3] | MAF European (dbVar)[3] |
|---|---|---|---|---|---|---|---|---|---|---|---|---|
| **Potentially Pathogenic SV (PP-SV)** | | | | | | | | | | | | |
| SLC3A1 | IED | chr2 | 44281377 | chr2 | 44281612 | DUP | L-pathogenic | 0.01 [4] | 0 | 0 | 0.0075 | 1.3e-04 |
| OCA2 | pLoF | chr15 | 28017719 | chr15 | 28020677 | DEL | L-pathogenic | 0.004 | 0.02 | 0 | 0.0015 | 0.001 |
| PIGN | pLoF | chr18 | 62152637 | chr18 | 62157701 | DEL | Pathogenic | 0.004 | 0 | 0 | 0.0013 | 1.3e-04 |
| SLC7A2 | pLoF | chr8 | 17418976 | chr8 | 17544122 | DEL | In dbVar | 0.009 | 0 | 0 | 0.003 | 0 |
| DNAJC15 | pLoF | chr13 | 43078470 | chr13 | 43079390 | DEL | In dbVar | 0 | 0 | 0.009 | 0 | 1.0e-04 |
| BCL2L11 | pLoF | chr2 | 111122626 | chr2 | 111125901 | DEL | This study | 0.005 | 0 | 0 | NA | NA |
| BARD1 | pLoF | chr2 | 214768022 | chr2 | 214772899 | DEL | This study | 0.005 | 0 | 0 | NA | NA |
| COL4A2/ COL4A1 | CG | chr13 | 110294204 | chr13 | 110633815 | DUP | In dbVar | 0.005 | 0 | 0 | 1.3e-04 | 6.3e-06 |
| SLC2A5 | IED | chr1 | 9045605 | chr1 | 9049441 | DUP | In dbVar | 0 | 0 | 0.009 | 7.3e-04 | 0.002 |
| FOXP1 | pLoF | chr3 | 71097066 | chr3 | 74525618 | INV | This study | 0.009 | 0 | 0 | NA | NA |
| WASF1 | pLoF | chr6 | 108167886 | chr6 | 110172775 | INV | In dbVar | 0.004 | 0 | 0 | 9.6e-05 | 0 |
| MLH1 | pLoF | chr3 | 37000362 | chr3 | 39352689 | INV | In dbVar | 0.004 | 0 | 0 | 4e-04 | 6.4e-06 |
| RB1 | pLoF | chr13 | 48466588 | chr13 | 48473911 | INV | In dbVar | 0.004 | 0 | 0 | 1.8e-04 | 1.3e-05 |
| CTNNA1 | pLoF | chr5 | 138903881 | chr19 | 21614900 | TRA | This study | 0 | 0 | 0.009 | NA | NA |
| AK8-DST | pLoF | chr9 | 132876361 | chr6 | 56896165 | TRA | This study | 0 | 0 | 0.009 | NA | NA |
| **PP-SV candidates classified as 'cautionary'** | | | | | | | | | | | | |
| LTBP1/BIRC6 | CG | chr2 | 32403832 | chr2 | 33107415 | DUP | In dbVar | 0 | 0 | 0.009 | 1.0e-04 | 0.0018 |
| PHC3-PRKACA | pLoF | chr3 | 1700090742 | chr19 | 14110142 | TRA | This study | 0.004 | 0 | 0 | NA | NA |
| KCTD3-DST | pLoF | chr1 | 215557414 | chr6 | 56652607 | TRA | This study | 0.009 | 0 | 0 | NA | NA |
| PKHD1 | pLoF | chr6 | 51981375 | chr15 | 30874073 | TRA | This study | 0.009 | 0 | 0 | NA | NA |

CG copy gain, chr chromosome, DEL deletion, DUP duplication, IED intragenic exon duplication, INV inversion, L-pathogenic Likely pathogenic, MAF minor allele frequency, pLoF potentially loss-of-function, pos position, TRA translocation. Note, all gene names are in italics.

[1] Gene impact type based on gene annotation.

[2] Population-matched Southern African non-cancer control cohort (n = 49).

[3] The ancestry-related MAF in dbVar were based on gnomAD[32] or TOPMed[42] SV study. The details of all dbVar studies (dbVar study name and ID) and reported allele frequencies were shown in Supplementary Data 1.

[4] Presenting at low-frequency rather than rare variants within the ancestrally-defined patient cohort.

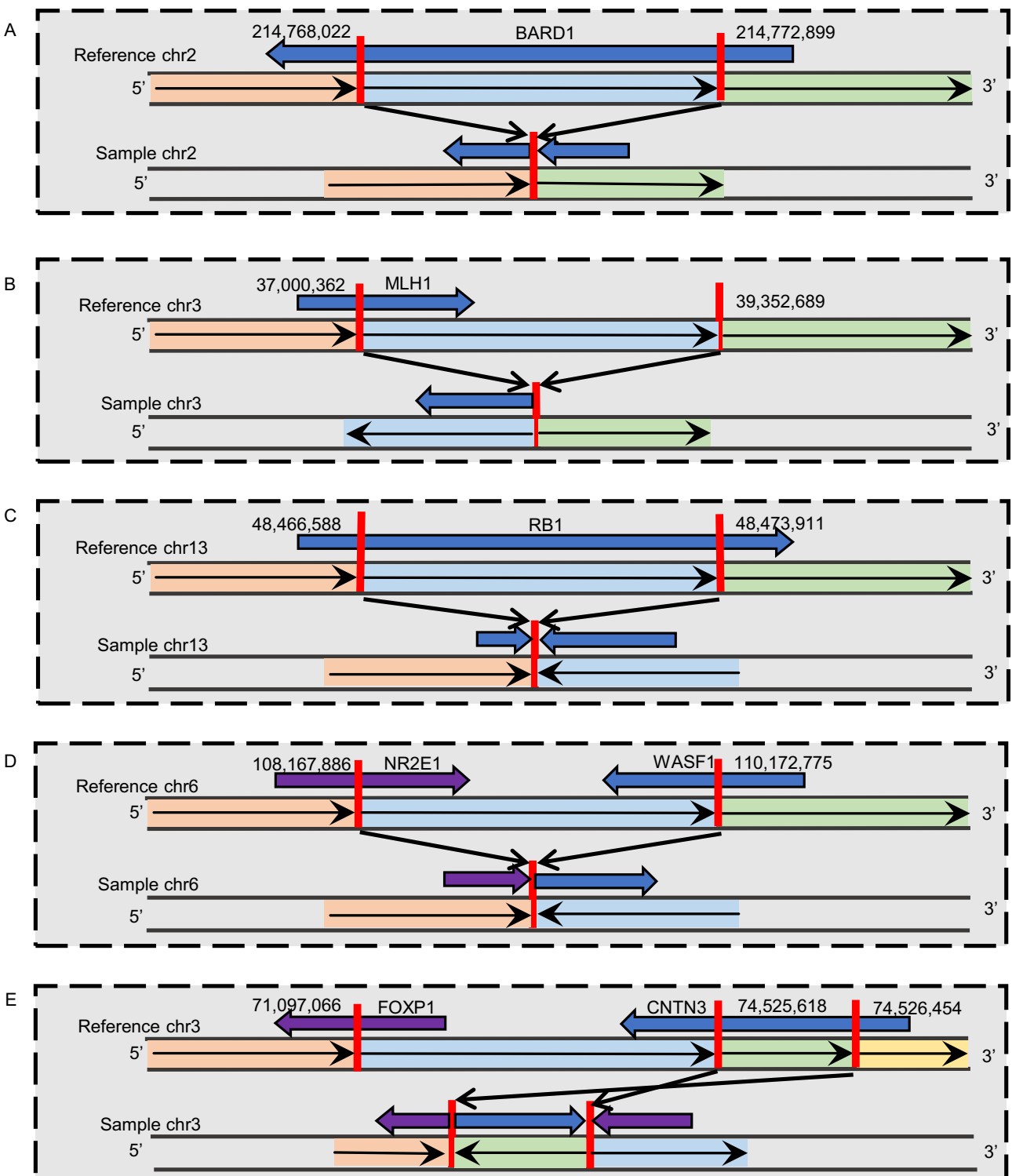

**Fig. 2 | African-specific PP-SVs disrupting well-known pathogenic cancer genes and/or PCa tumour suppressor genes, including DNA damage response genes. A** 4877 base pLoF deletion on DNA damage repair gene *BARD1*. **B** pLoF INV impacting PCa DNA mismatch repair gene *MLH1*. **C** pLoF INV impacting PCa tumour suppressor *RB1*. **D** pLoF INV impacting PCa tumour suppressor gene *WASF1*. **E** pLoF INV impacting PCa tumour suppressor gene *FOXP1*. More details of SV region and/ or breakpoints on impacted genes and visual inspection of sequencing reads using Integrative Genomic Viewer[51] are shown in Supplementary Figs. 8, 12, 13, 14 and 15, respectively.

(Supplementary Fig. 5) and a *DNAJC15* 920 bp DEL in a European patient (Supplementary Fig. 6). Another two identified PP-SVs are unknown pLoF DELs, which identified in a single African patient each, including a *BCL2L11* 3,275 bp (Supplementary Fig. 7) and DNA damage repair gene *BARD1* 4877 bp DEL (Fig. 2A and Supplementary Fig. 8).

Of the two dbVar whole-gene DUPs, the *COL4A2* 339,611 bp CG, with breakpoints disrupting *COL4A1* and *NAXD*, observed in a single African patient is defined as a PP-SV (Supplementary Fig. 9), as *COL4A2* indicating oncogenic behaviour in gastric and breast cancers (Supplementary Table 7). In contrast, the *TTC27* 703,583 bp DUP observed

in a single European patient is afforded 'cautionary' PP-SV status (Supplementary Fig. 10). Although *TTC27* is absent in three cancer gene databases, the breakpoints disrupt MSigDB and COSMIC CGC genes *BIRC6* and *LTBP1*, resulting in a *LTBP1-BIRC6* gene fusion of unclear effect. Observed in a single European patient, a 3836 base DUP directly impacts exon 4 of *SLC2A5* (Supplementary Fig. 11), which downregulated in PCa ($p$-value < 0.001, GENT2 T-test) and has been identified as an oncogenic behaviour (Supplementary Table 7), therefore allocated PP-SV status.

Of the five pLoF INVs, those impacting *MLH1*, *RB1* and *WASF1* are in dbVar, while *FOXP1* and *NSD3* INVs are unknown. As *NSD3* has been identified as oncogenic in multiple cancers, the associated INV is classified here as unlikely pathogenic, with all remaining pLoF INVs classified as PP-SVs, as they disrupting known to PCa and Lynch Syndrome predisposing DNA mismatch repair gene *MLH1* and PCa tumour suppressor genes *RB1*, *WASF1*, and *FOXP1* (Supplementary Table 7). Identified in a single African patient each (Supplementary Fig. 12–14), the three dbVar INVs were reported as rare by the recent TOPMed SV study[42], in which *WASF1* INV was also identified as African-specific (Table 1 and Supplementary Data 1). The unknown INV impacting *FOXP1* was identified in two African patients (Fig. 2E and Supplementary Fig. 15).

Of the 10 pLoF TRAs, five impacting genes of *GRM8*, *WDR43*, *NPM1*, *NUSAP1* and *MECOM* with oncogenic properties (Supplementary Table 7), therefore, are classified as unlikely pathogenic. *PKHD1* TRA identified in two African patients received a 'cautionary' PP-SV classification, as identified as potential oncogenic in colon cancer, while potential tumour suppressor in colorectal cancer (Supplementary Table 7). As *CTNNA1* was known to have tumour suppressor behaviour across multiple tumour types (Supplementary Table 7), here we classify the European-specific pLoF *CTNNA1* TRA as a PP-SV (Supplementary Fig. 16). The remaining pLoF TRAs result in *PHC3-PRKACA* (1 African patient), *KCTD3-DST* (2 African patients) and *AK8-DST* (1 European patient, Supplementary Fig. 17) as current unknown gene fusions. *PHC3-PRKACA* was classified as 'cautionary' PP-SV, as *PHC3* showed a potential cancer suppressor effect in PCa, while *PRKACA* appears to portray oncogenic behaviour (Supplementary Table 7). Although unknown to PCa, both *DST* and *AK8* have demonstrated tumour suppressor behaviour, conversely, *KCTD3* with an unclear role in cancer (Supplementary Table 7). Here we classify *AK8-DST* as a PP-SV, while *KCTD3-DST* is assigned 'cautionary' PP-SV status.

## PP-SVs associated with clinical characters, expression and tumour features of causality

The clinicopathological features of the study cohort has been previously described[18,28]. In brief, African patients show a 5-year greater mean age and 25-fold greater PSA level at diagnosis compared to European patients (Supplementary Table 1). Based on our previous observations[20], high-risk or aggressive PCa were defined as ISUP GG3 and conversely, low-risk disease presentation as ISUP GG<3. Biased towards aggressive disease presentation (82% African, 86.0% European), it was notable that all four patients with a pathogenic or likely pathogenic SVs presented with the aggressive disease at diagnosis, 92.9% (13/14) of PP-SV and 83.3% (5/6) cautionary PP-SV presenting patients (Table 2). We further concur that all SVs are likely germline as their variant allele frequencies (VAFs) are as expected for heterozygous inheritance (average of 41.8%; range 30% to 51%, Supplementary Table 8). Due to the lack of available Southern African population-matched non-cancerous data, here we further interrogated 49 younger-aged non-cancerous Southern Africans for potentially pathogenic SVs within the 19 PP-SV-defined candidate genes. Besides the previously discussed *OCA2* ClinVar defined "likely pathogenic" SV, no PP-SVs were identified within this cohort (Table 1). In turn, additional non-matched pLoF SVs were identified impacting *OCA2*, *BARD1* and *SLC2A2* (Supplementary Table 3).

In parallel, blood was available for 20 patients including five with a PP-SV allowing for further experimental expression analyses through RNA-sequencing interrogation. Consequently, the unknown pLoF PP-SVs impacting *BCL2L11* (KAL0101) and *FOXP1* (UP2101) and the known *MLH1* pLoF PP-SV (SMU080) showed reduced expression levels – with $\log_2$ fold change values of -1.36, -1.07 and -0.67, respectively (Supplementary Fig. 18A). Additionally, high expression of *COL4A2* in contrast to lack of expression of *COL4A1* in patient UP2039 (Supplementary Fig. 18B) concurs with our expected copy gain impact as a consequence of the former gene duplication (Supplementary Fig. 9). The near to average expression of 'cautionary' PP-SV impacting *PHC3* and *PRKACA* in patient SMU061, with normalised counts of 5863.56 *vs.* 6816.17 and 1689.19 *vs.* 1569.10, respectively (Supplementary Fig. 18C), provides further validation for our cautionary classification.

Furthermore, tumour-matched samples from PP-SV presenting patients were assessed for biallelic loss or a second somatic hit, both potential indicators of gene-relevant causality[49]. We used TITAN to infer for copy number (CN) loss as a prediction of loss of heterozygosity (LOH-CNL) due to deletion of wild-type allele, and CN neutral or gain LOH (LOH-CNN and LOH-CNG, respectively) due to accompanied duplication of the mutant allele[50]. After correction for tumour purity and ploidy (Supplementary Data 3), we identified LOH in five PP-SV presenting patients, including two with ClinVar validated pathogenic or likely pathogenic SVs and three presenting with ISUP 4-5 disease (Table 2). LOH-CNL was observed in a single patient SMU083 resulting in biallelic loss of *PIGN*. LOH-CNG was observed in two patients, indicating wild-type allele loss for *SLC3A1* and *SLC7A2* in patients SMU094 and KAL0054, respectively. LOH-CNN, indicating wild-type allele loss with amplification of tumour suppressor losses, was observed in *BCL2L11* and *BARD1* for patients KAL0101 and N0073.

Overall and irrespective of ancestry, patients with germline PP-SVs (14 African, 4 European) showed less oncogenic driver variants than non-PP-SV presenting cases (99 African, 53 European), although not statistically significant (250, range: 101-437 *vs* 316, range: 105–772). In addition, we found the same gene second hit by somatic CN alterations including CN loss impacting *SLC3A1*, *SLC7A2* and two cautionary fusion PP-SVs in in an African patient each (all ISUP GG ≥ 3 at diagnosis) and in turn CN gain impacting *OCA2*, *FOXP1* and *WASF1* in each of three African patients with advanced ISUP GG5 disease (Supplementary Table 9 and Table 2). Notably, the single European patient presenting with the cautionary PP-SV impacting *LTBP1*, with second hit somatic CN gain, underwent surgery at 54 years of age for ISUP GG5 disease.

## Discussion

ClinVar defined pathogenic (or likely pathogenic) SVs disrupting solute carrier family 3 member 1 (*SLC3A1*), OCA2 melanosomal transmembrane protein (*OCA2*) or phosphatidylinositol glycan anchor biosynthesis class N (*PIGN*) were observed in 3.5% (4/113) of African patients. Specifically, the *SLC3A1* intragenic exon DUP was identified in two patients presenting with ISUP GG4, while the *OCA2* and *PIGN* pLoF DELs presented in a single patient, each with ISUP GG5 and ISUP GG3 PCa, respectively (Table 2). Visually inspecting the three PP-SVs using Integrative Genomic Viewer[51], *SLC3A1* DUP was found with three supporting read-pairs in sample N0001 (Supplementary Fig. 3), and split-reads and more than 40% increase in read depth comparing to ±10 kb of the SV region in both samples (Supplementary Table 10), while *OCA2* and *PIGN* DELs were found with 16 and 6 supporting read-pairs respectively (Supplementary Fig. 1, 2), and have 44-51% reduction in read depth (Supplementary Table 10). *SLC3A1* is an amino acid transporter, which, through heterodimerisation with *SLC7A9* is responsible for cystine reabsorption through cationic and neutral amino acid exchange[52]. Mutations, including SVs, in *SLC3A1* are associated with cystinuria, an inherited disease that results in the formation of cystine stones in the kidney, with disease presentation suggested to require biallelic loss[53]. *SCL3A1* over-expression has been associated with

**Table 2 | Clinicopathological features, variant allele frequencies (VAFs), associated somatic loss of heterozygosity (LOH) and/or second gene hit of prostate cancer (PCa) patients by ancestry presenting with potentially pathogenic (PP) SVs and cautionary PP-SVs as defined by this study**

| Gene name | Pathogenicity | SV (impact) type | Patient No. | Patient ID[2] | Expression[1] | VAF[3] | Ancestry | Age | PSA | ISUP GG | Tumour LOH[4] | Tumour 2nd hit[5] |
|---|---|---|---|---|---|---|---|---|---|---|---|---|
| SLC3A1 | PP-SV (LP) | DUP (IED) | 1 | N0001 | ND | 0.33 | African | 75 | 22.9 | 4 | LOH-neg | No |
| – | – | | 2 | SMU094 | ND | 0.30 | African | 64 | 15 | 4 | LOH-CNG | CNL |
| OCA2 | PP-SV (LP) | DEL (pLoF) | 1 | N0059 | ND | 0.45 | African | 79 | 153 | 5 | LOH-neg | CNG |
| PIGN | PP-SV (P) | DEL (pLoF) | 1 | SMU083 | ND | 0.51 | African | 86 | 40.5 | 3 | LOH-CNL | No |
| SLC7A2 | PP-SV | DEL (pLoF) | 1 | UP2035 | ND | 0.48 | African | 70 | 680 | 5 | LOH-neg | CNL |
| – | – | | 2 | KAL0054 | ND | 0.49 | African | 64 | 42.9 | 5 | LOH-CNG | No |
| DNAJC15 | PP-SV | DEL (pLoF) | 1 | 17135 | ND | 0.48 | European | 63 | 7.8 | 5 | LOH-neg | No |
| BCL2L11 | PP-SV | DEL (pLoF) | 1 | KAL0101 | Low | 0.45 | African | 71 | 32.3 | 5 | LOH-CNN | No |
| BARD1 | PP-SV | DEL (pLoF) | 1 | N0073 | ND | 0.49 | African | 62 | unk | unk | LOH-CNN | No |
| COL4A2/ COL4A1 | PP-SV | DUP (CG) | 1 | UP2039 | High/NE | 0.35 | African | 71 | 319 | 4 | LOH-neg | No |
| SLC2A5 | PP-SV | DUP (IED) | 1 | 11099 | ND | 0.33 | European | 70 | 9.9 | 5 | LOH-neg | No |
| FOXP1 | PP-SV | INV (pLoF) | 1 | UP2101 | Low | 0.41 | African | 57 | 75 | 5 | LOH-neg | CNG |
| – | – | – | 2 | N0084 | ND | 0.41 | African | 65 | 591 | 4 | LOH-neg | No |
| WASF1 | PP-SV | INV (pLoF) | 1 | N0048 | ND | 0.32 | African | 70 | 83.3 | 5 | LOH-neg | CNG |
| MLH1 | PP-SV | INV (pLoF) | 1 | SMU080 | Low | 0.43 | African | 64 | 23.3 | 4 | LOH-neg | No |
| RB1 | PP-SV | INV (pLoF) | 1 | SMU064 | ND | 0.37 | African | 70 | 13.7 | 3 | LOH-neg | No |
| CTNNA1 | PP-SV | TRA (pLoF) | 1 | 13179 | ND | 0.50 | European | 59 | 8.4 | 5 | LOH-neg | No |
| AK8-DST | PP-SV | TRA (pLoF) | 1 | 11452 | ND | 0.47 | European | 67 | 11 | 1 | LOH-neg | No |
| LTBP1/ BIRC6 | Cautionary PP-SV | DUP (CG) | 1 | 5287 | ND | 0.39 | European | 54 | 4.3 | 5 | LOH-neg | CNG |
| PHC3-PRKACA | Cautionary PP-SV | TRA (pLoF) | 1 | SMU061 | Avg./Avg. | 0.49 | African | 65 | 12.1 | 3 | LOH-neg | CNL |
| KCTD3-DST | Cautionary PP-SV | TRA (pLoF) | 1 | UP2039 | ND | 0.40 | African | 71 | 319 | 4 | LOH-neg | CNL |
| – | – | – | 2 | SMU101 | ND | 0.42 | African | 70 | 4.3 | 3 | LOH-neg | No |
| PKHD1 | Cautionary PP-SV | TRA (pLoF) | 1 | N0056 | ND | 0.36 | African | 70 | 153 | 5 | LOH-neg | CNG |
| | | | 2 | SMU196 | ND | 0.39 | African | 47 | 9.5 | 1 | LOH-neg | No |

*CG* copy gain, *CNG* copy-number gain, *CNL* copy-number loss, *CNN* copy-number neutral, *DEL* deletion, *DUP* duplication, *IED* intragenic exon duplication, *INV* inversion, *ISUP GG* International Society of Urological Pathology Group Grading, *TRA* translocation, *LOH* loss of heterozygosity, *LP* likely pathogenic, *ND* not determined, *NE* no expression, *neg* negative, *P* pathogenic, *pLoF* potentially loss-of-function, *SV* structural variant, *unk* unknown, *VAF* variant allele frequency. Note, all gene names are in italic; age is in years and PSA in ng/mL. Pathogenic (P) and Likely pathogenic (LP) are ClinVar defined.
[1]Gene expression derived from blood-matched RNAseq analysis. [2]While no known family history of prostate, breast or ovarian cancer was reported for these patients, SMU080 reported a sister with cervical cancer and SMU061 a mother with stomach cancer. [3]Variant allele frequency (Supplementary Table 8). [4]Loss of heterozygosity status was inferred from TITAN. [5]The details of the second somatic hit locations were shown in Supplementary Table 9.

enhanced tumourigenesis in breast cancer while blocking *SCL3A1* has suggestive therapeutic potential[39]. Taken together, it is notable that somatic LOH-CNG, with second hit CN loss, was observed for the younger of the two *SLC3A1* PP-SV presenting African patients (SMU094). *OCA2* is a pigmentation gene with inherited mutations associated with oculocutaneous albinism[54]. Polymorphisms have been associated with skin cancers[55], as well as clinical response and survival in breast cancer patients having received neoadjuvant chemotherapy[56]. Notably, the ISUP GG5 presenting pLoF germline SV African patient also with a second hit somatic *OCA2* CN gain. Inherited *PIGN* mutations have been associated with multiple congenital anomalies-hypotonia-seizures syndrome and Fryns syndrome, with some mutations related to milder forms of clinical presentation[57,58]. *PIGN* is involved in the biosynthesis of glycosylphosphatidylinositol, which has been shown to suppress cancer chromosomal instability[37]

through PIGN complexed spindle assembly checkpoint regulation[38], a common phenomenon in solid tumours[59]. Biallelic *PIGN* loss is tentatively predicted in the tumour of the older aged presenting African patient. While assumed pathogenic, an association between *SCL3A1*, *OCA2* or *PIGN* mutation and PCa is yet to be elucidated.

As our study is biased towards under-represented African patients, it is highly plausible that the majority of SVs detected are unlikely to be represented in ClinVar. As such, it is critical that we develop a best-fit workflow for PP-SV prediction. The four SV impact prediction tools used in this study were chosen based on the criteria of easy-to-use (either web-based or packed as software), providing pathogenicity scores or labels, accepting multiple SVs and covering all SV types. However, there are multiple factors to be taken into consideration when using SV impact prediction tools to establish potential pathogenicity, as different tools have limitations in applicable SV

types, regions or diseases, as well as different scoring systems. While all tools can predict the impact of DELs and DUPs, StrVCTVRE is limited to DELs and DUPs in exonic regions. Besides predicting the simpler SVs, CADD-SV is capable of annotating INSs, POSTRE annotates INVs and TRAs, while PhenoSV is able to predict the impact of all these three types. POSTRE doesn't work for all diseases or phenotypes. Therefore, combining multiple tools is necessary to cover all SV types and increase the confidence level. Another factor is the choice of threshold to establish pathogenicity. POSTRE and PhenoSV define the threshold of pathogenicity, but StrVCTVRE and CADD-SV are limited to scores and calling for thresholds to be established depending on individual study aims. In this study, we have decided the thresholds based on tools' validated results from the database (90% sensitivity in ClinVar by StrVCTVRE[43] and top 10% in gnomAD by CADD-SV[44]). When combining results from multiple tools, we found the requirement of passing thresholds of all four tools identified two PP-SV candidates (out of 1843 SVs) (Supplementary Table 4), with notable failure to identify the three ClinVar pathogenic/likely pathogenic SVs (Supplementary Table 6). As such, PP-SV candidate classification in this study required an SV to pass thresholds of at least two impact prediction tools, with disrupted genes requiring further clarification as hallmarks or drivers in cancer gene databases (MSigDB and COSMIC CGC).

Using our described workflow, 12 SVs were predicted as PP-SVs, identified in 7.0% (4/57) of European and 8.8% (10/113) of African patients, bringing the total of African patients presenting with a potential pathogenic SV to 12.4% (14/113). Remarkably, five of our African-specific PP-SVs included well-known pathogenic cancer genes and/or PCa tumour suppressor genes, including DNA damage response genes. Most notably, the DNA mismatch repair tumour suppressor gene mutL homologue 1 (*MLH1*) commonly mutated in Lynch Syndrome, including cases with PCa[60], is a known candidate gene in PCa germline testing panels[20]. While PCa patients presenting with pathogenic *MLH1* mutations were reported to have significantly higher disease burden for African Americans[24], here we found a dbVar known *MLH1* pLoF INV with around 11 supporting short read-pairs (Supplementary Fig. 12) in a 64-year-old African male presenting with ISUP GG4 at diagnosis and at the time of diagnosis no somatic LOH or a second gene hit. Not recognised as a PCa germline testing panel gene, forkhead box P1 (*FOXP1*) is an established PCa tumour suppressor driver gene, with CN loss increasing cell proliferation and migration, and poor prognosis[61]. Recently, we showed *FOXP1* to be equally impacted by predominantly CN loss in African compared with European-derived tumours (20% of 183 tumours)[18]. Here we found a germline inverted duplication impacting *FOXP1* with around 18 supporting read-pairs in two African patients (Supplementary Fig. 15). Notably, one African patient (UP2101) presented 10 years earlier than the cohort average receiving an ISUP GG5 diagnosis, however, this patient also presented with a second hit somatic *FOXP1* CN gain. Loss of the *BRAC1*-associated RING domain-1 (*BARD1*) DNA damage repair gene has been found to induce homologous recombination deficiency and increase the sensitivity to PARP inhibitor in PCa cell lines[62]. Here the unknown *BARD1* exon 5 DEL, supported by 10 read-pairs and with around 50% reduction in read depth comparing to ±10 kb of the DEL region (Supplementary Fig. 8 and Supplementary Table 10), was identified in a 62-year-old African PCa patient with unknown pathology, with associated amplification of the deleted allele during tumourigenesis. While a paediatric cancer predisposing tumour suppressor gene commonly mutated in retinoblastoma and to a lesser extent osteosarcoma[63], and less common as an adult cancer predisposing gene[64], Retinoblastoma transcriptional corepressor 1 (*RB1*) is recognised as one of five most prevalent somatically mutated genes in metastatic cancers[65], with *RB1* loss in prostate tumours associated with poor patient outcomes[66]. To the best of our knowledge, no germline potentially pathogenic *RB1* variant has been reported for PCa, which includes a pLoF INV of exon 24 with three supporting read-pairs

(Supplementary Fig. 13) in a single ISUP GG3 diagnosed African patient. Lastly, we found a PP-SV in the tumour suppressor gene WASP family member 1 (*WASF1*) with loss previously associated with aggressive or metastatic lethal PCa[67]. A potentially pathogenic INV, previously reported at MAF of 9.6e-05 in Africans and resulting in *NR2E1-WASF1* fusion, was identified in a single African patient presenting at 70 years of age with ISUP GG5 PCa, showed 14 supporting read-pairs (Supplementary Fig. 14), with somatic hyper-amplification during tumourigenesis.

Other notable PP-SV DELs impacting tumour suppressor genes unknown to PCa include solute carrier family 7 member 2 (*SLC7A2*) and DnaJ heat shock protein family member C15 (*DNAJC15*). Knockdown of *SLC7A2* has been shown to promote viability, invasion and migration of ovarian cancer[68] and enhance proliferation of non-small-cell lung cancer cells[69], while *DNAJC15* has tumour suppressor behaviour in breast cancer[70]. Identified in two African patients presenting with ISUP GG5 disease, loss of *SLC7A2* exons 1 and 2, supported by 10 read-pairs and with around 50% reduction in read depth (Supplementary Fig. 5 and Supplementary Table 10) has previously been reported in African populations at MAF of 0.03 (Supplementary Data 1). Notably, the younger of the two patients showed loss of the wild-type allele during tumourigenesis. Specific to Europeans (MAF = 1.0e-04), loss of *DNAJC5* exon 4 supported by 13 read-pairs and with around 50% reduction in read depth (Supplementary Fig. 6 and Supplementary Table 10), was identified in a single European patient presenting for surgery at age 63 years with ISUP GG5 disease, yet appears to remain diploid during tumour development. While not associated with PCa, the loss of BCL2 like 11 (*BCL2L11*) and catenin alpha 1 (*CTNNA1*) has been identified as drivers of tumourigenesis and promoting invasion and metastasis of multiple cancers[71,72]. Unknown SVs include; *BCL2L11* pLoF DEL impacting exon 2 with more than 20 supporting read-pairs and with around 50% reduction in read depth (Supplementary Fig. 7 and Supplementary Table 10) was identified in a single African patient presenting at age 71 years with ISUP GG5 PCa, while pLoF TRA interrupting *CTNNA1* with more than 20 supporting read-pairs (Supplementary Fig. 16) in a single European patient presenting at age 59 years with ISUP GG5 PCa. Notably, the African *BCL1L11* PP-SV presenting patient showed further somatic LOH. Another currently unknown SV identified included the potentially pathogenic inter-chromosomal TRA with around 18 supporting read-pairs (Supplementary Fig. 17) leading to an adenylate kinase 8 (*AK8*)- dystonin (*DST*) fusion in a single European patient (ISUP GG1, 67 years). Although no associations have been made between PCa, higher expression of *DST* has been identified to promote pathogenesis and development of breast cancer, while *AK8* downregulation has been found to promote migration and invasion of uterine carcinosarcoma[73].

Two identified PP-SVs have the potential to increase gene dosage of well-known oncogenes collagen type IV alpha 2 chain (*COL4A2*) and solute carrier family 2 member 5 (*SLC2A5*) through whole-gene and intra-genic exon duplication, respectively. Although not associated with PCa, *COL4A2* loss has been identified to inhibit triple-negative breast cancer cell proliferation and migration[74] and its mutations as a risk factor for familial cerebrovascular disease[75], while inactivation of *SLC2A5* has been found to inhibit cell proliferation and migration in multiple cancer cell lines[76]. The whole *COL4A2* DUP with more than 20 supporting read-pairs and more than 50% gain in read depth (Supplementary Fig. 9 and Supplementary Table 10) was identified in a single African patient (ISUP GG4, 71 years) and the exon 4 DUP in *SLC2A5* with more than 20 supporting read-pairs and more than 50% increase in read depth (Supplementary Fig. 11 and Supplementary Table 10) was identified in a single European patient (ISUP GG5, 70 years). No LOH or second hits were observed during tumorigenesis.

Using short-read sequencing data for SV calling and genotyping remains a potential limitation, appreciating that SVs in difficult-to-sequence regions may have been overlooked[77]. To ensure the highest

possible accuracy of SV detection and population allele frequency estimation, we required high-confidence calls from two SV callers and high-quality genotype calls at both the population- and individual-level, while all PP-SVs were visually inspected. Due to a lack of available expression data, we were unable to validate the direct impact of identified PP-SVs and cautionary PP-SVs. While LOH or second hits in the developing tumours added further possible causality to several candidate genes, one cannot ignore that LOH can occur by chance. Conversely, we cannot exclude hypermethylation inactivation as a second hit for the remaining gene candidates. Additional study limitations related to our southern African cohort include (i) a lack of population-matched healthy controls or regionally relevant population-wide whole genome data, (ii) the on average older age at PCa presentation[23,78] and lack of PCa knowledge as related to family history[79], both criteria traditionally used for genetic testing, and (iii) a lack of African-relevant data in currently pathogenicity prediction databases. As a consequence of these illimitations, our study calls for further African-inclusive efforts and for the establishment of guidelines for pathogenic SV identification using both short and/or long-read sequencing approaches, making these methods accessible for routine multi-ancestral germline testing.

Here, we provide substantial evidence that inherited SVs may not only be contributing to PCa pathogenicity but also associated ancestry disparity. We observed three ClinVar-defined pathogenic or likely pathogenic PP-SVs (*SLC3A1*, *OCA2* and *PIGN*) and 12 predicted PP-SVs, including seven known SVs (*SLC7A2, DNAJC15, COL4A2, SLC2A5, WASF1, MLH1* and *RB1*), and five unknown SVs (*BCL2L11, BARD1, FOXP1, CTNNA1* and *AK8-DST*) of which patients presenting with *BCL2L11* and *FOXP1* SVs show associated loss of gene expression, suggesting that inherited SVs may constitute an under-appreciated contribution to PCa pathogenicity. Furthermore, the identification of African-private (8 known, 3 unknown) and European-private (2 known, 2 unknown) PP-SVs allows for further speculation with regard to associated racial disparities while improving the detection rate for PCa germline testing with SV inclusivity, and in turn raising limitations for African inclusion and associated clinical care.

## Methods

### Participant recruitment and ethics approval

Irrespective of country of origin, all individuals provided written and signed informed consent to participate in the study and publish. Conforming to the principles of the Helsinki Declaration, South African patients were recruited as part of the Southern African Prostate Cancer Study (SAPCS) with approval granted by the University of Pretoria Faculty of Health Sciences Research Ethics Committee (HREC, with US Federal-wide assurance FWA00002567 and IRB00002235 IORG0001762; #43/2010). In Australia, participant recruitment was approved by the St Vincent's HREC (#SVH/12/231). As all patients underwent a prostate biopsy (South Africa) or surgery (Australia), all are assumed to be biologically male. Samples were shipped to the Garvan Institute of Medical Research and, subsequently the University of Sydney in accordance with institutional Material Transfer Agreements (MTAs), as well as additional Republic of South Africa Department of Health Export Permit (National Health Act 2003; J1/2/4/2 #1/12). This study was approved by the St. Vincent's HREC (#SVH/15/227) for genomic interrogation. Additional IRB review and approval for genomic interrogation was granted by the Human Research Protection Office of the US Army Medical Research and Development Command E02371 (TARGET Africa) and E03280 (HEROIC PCaPH Africa1K).

### WGS data generation

To avoid technical and analytical biases, all samples (whole blood) were processed (beginning at DNA extraction), data generated and analysed within a single laboratory using a single computational pipeline[18,28]. In brief, whole-genome sequencing data were generated using Illumina HiSeq X Ten (21 cases) or NovoSeq (149 cases) instruments with 2 × 150 cycle paired-end mode at the Kinghorn Centre for Clinical Genomics (Garvan Institute of Medical Research, Australia). Following the BROAD's best practice recommendations for "data preprocessing for variant discovery", sequencing reads were aligned to GRCh38 reference genome with alternative contigs using scalable FASTQ-to-BAM (v2.0) workflow with default settings[80]. The mean depth of coverage for all samples were 45.9X (range 30.2–97.6X).

### Structural variant calling and high-confidence SV filtering

Germline SVs were called using Manta (v1.6.0)[81] and GRIDSS (v2.13.3)[82,83]. SV types reported by Manta included DEL, tandem DUP, INS and adjacent breakends (BNDs) for a fusion junction with an inverted sequence or in an inter-chromosomal rearranged genome. Pairs of BND in inverted junctions were annotated as inversions (INV). Pairs of BND in different chromosomes were annotated as inter-chromosomal translocations (TRA). Conversely, GRIDSS reports BND for all fusion junctions resulting from any SV event. Simple SV types, defined as DEL, DUP, INS, INV and TRA, were assigned based on the strands and ALT field in VCF (modified from GRIDSS accompanied R script: simple-event-annotation.R).To obtain a high-confidence SV call set, we integrated call sets from Manta and GRIDSS and generated a concordant call set for each genome. Two SV calls were considered as concordant if they were reported as "PASS" by one of the two callers and have matching SV type and reported breakpoint positions within 200 bp of each other. *Bedtools pairtopair*[84] was used to compare two call sets.

### Population-level genotyping and high-confidence genotype call filtering

We used Graphtyper2 (v2.7.5)[85] to re-genotype SVs for all samples. Following published guidelines, we merged all high-confidence SV sets persample (individual VCFs) using svimmer (https://github.com/DecodeGenetics/svimmer) with default parameters. The individual VCFs were in the format of Manta VCFs, as Manta provides detailed information on the exact breakpoint sequence, which is the essential information required by Graphtyper2. We extracted all SVs with the "aggregate" model as suggested and obtained 57,096 SVs with "PASS" in the FILTER field in VCF. We also required SVs to have more than 50% of genotype calls as "PASS" (PASS_ratio $\geq$ 0.5 in INFO field), resulting in 42,966 SVs.

To further filter SV genotype calls on a per-sample basis, we set the SV genotype as missing if the genotype filter tag (FT) is not "PASS" for all SVs, except BND. For BND, as the FT tag is not available, we set the BND genotype with genotype quality (GQ) < 20 as missing. We then excluded SVs with genotype missingness rate > 20% in either African or European genomes, resulting in 33,340 SVs. We further removed 97 SVs with an allele frequency of 100%, indicating the difference of the sample genomes to reference genome. The allele frequency of each SV was then calculated based on the high-quality genotype calls only.

### Gene annotation and functional impact of SVs

All SVs were annotated against gene regions from the Ensembl human gene annotation file (GRCh38 assembly, release 108)[86]. As multiple transcripts can be available for a single gene, the Ensembl Canonical transcript was used (http://www.ensembl.org/info/genome/genebuild/canonical.html). By comparing the position of SV breakpoint with gene regions using bedtools[84], we examined nine gene overlapping categories with gnomAD[32], including potential Loss of Function (pLoF), Copy Gain (CG), Intragenic Exon DUP (IED), partial gene DUP, whole-gene INV, UTR SVs, promoter SVs, intronic SVs and intergenic SVs. In addition, we defined partial-exon DUP as both breakpoints contained within the same gene, while neither both within exons (pLoF) nor fully overlapped at least one exon (IED). Promoters were defined as a 1 kb window before each transcription start site on

the transcribed strand. We labelled SVs as enhancer-disruptive if at least one breakpoint was contained within a gene's enhancer, by comparing to GeneHancer[87] regulatory elements regions. GeneHancer regulatory elements and gene interactions "double elite" subset was downloaded from UCSC Table geneHancerInteractionsDoubleElite [last updated 15/01/2019] from GeneHancer track for GRCh38. The transcript structure plots were generated based on Ensembl human gene annotation (GRCh38 assembly, release 108) using R package ggtranscript (v0.99.3)[88]. The sequencing depth of DEL or DUP regions and their ±10 kb regions were calculated using samtools (v1.6) depth command[89].

Short-read data detect the SV signatures from aligned reads around the SV breakpoints, and is hard to capture the whole large SVs[90]. Therefore, we restricted the disrupted genes of SVs greater than 1Mbp to be genes overlapped by SV breakpoints for downstream analysis.

### Identification of dbVar concordance and unknnown SVs

The NCBI's database of human genomic structural variation (dbVar) [last updated 30/10/2023][91] were used to identify dbVar concordance and unknown SVs. The dbVar database included a total of 6,476,337 unique SVs, including 86,686 SVs with interpretations of their significance to disease in the ClinVar database[92]. Structural variants concordant to dbVar SVs were defined as having both breakpoints within 200 bases of dbVar-defined SV breakpoints. The ancestry-related variant allele frequency of SVs (Supplementary Data 1) were derived from dbVar pages of SVs or VCFs uploaded by different dbVar studies to dbVar's FTP site.

### Pathogenicity prediction

The pathogenicity of SVs were predicted through prediction tools StrVCTVRE[43], CADD-SV[44], POSTRE[45] and PhenoSV[46]. StrVCTVRE only scores the deleteriousness of DEL and DUP overlapping one or more exons, CADD-SV scores DEL, DUP and INS, POSTRE predicts the impact of DEL, DUP, INV and TRA, and PhenoSV works for all five SV types. As POSTRE only accepts genome coordinates on reference genome Hg19, the liftOver function from rtracklayer package in R was used to lift SV coordinates from Hg38 to Hg19. As suggested by StrVCTVRE, the ClinVar 90% sensitivity threshold (0.37) was used to define potentially pathogenic SVs. The scaled CADD-SV scores range from 0 (potentially benign) to 48 (potentially pathogenic), indicating the position of the input SV within the gnomAD-SV score distribution. The threshold of 10 for the CADD-SV score was used to establish potential pathogenicity, corresponding to the top 10% score observed in gnomAD-SV. The threshold of 0.8 and 0.5 for POSTRE and PhenoSV score, respectively, was used in this study, which is the threshold of pathogenicity labelling defined by POSTRE and PhenoSV.

The hallmark gene sets and oncogenic signature gene sets were downloaded from the Human Molecular Signature Database (MSigDB v2023.1)[47]. The MSigDB oncogenic signature gene sets included genes representing signatures of cellular pathways which are often dysregulated in cancer. Cancer-driver genes were downloaded from the COSMIC Cancer Gene Census (GRCh38 COSMIC v98, downloaded 26/09/2023).

### PP-SV allele frequency predictions

The VAF of each PP-SV was calculated as the fraction of altered sequencing reads within each SV region. Three types of altered sequencing reads were considered, including the reduced or increased sequencing reads due to DEL or DUP, discordantly aligned read-pairs and split reads. For DEL and DUP, the altered read count was calculated as the average read depth difference between the SV region and its ±10 kb regions. The total read count of DEL was measured as the average read depth of ±10 kbp region to SV region, and that for DUP was the average read depth of SV region. For INV and TRA, the altered

reads, including discordantly aligned reads and split-reads, within ±150 bp of each breakpoint were manually inspected using IGV. The total read count was measured as average sequencing read depth in ±150 bp region to INV or TRA breakpoints, calculated using samtools (v1.6) depth command. The SV VAF can be underestimated due to one read can show both discordant read-pair and split-reads, while only counted once. In addition, non-reference read-pairs is possible to be aligned properly in the SV region. In addition, whole genome data was made available for 49 Southern Bantu-matched disease-free individuals for further PP-SV candidate gene interrogation for comparative analyses. SV calling and genotyping, and high-confidence SV filtering were implemented exactly the same as PCa patients.

### Patient matched RNA-seq analysis for gene expression

Whole blood was available for 20 African patients (KAL047, KAL054, KAL061, KAL0101, N0053, N0056, SMU061, SMU080, SMU094, SMU109, SMU141, UP2035, UP2039, UP2092, UP2093, UP2100, UP2101, UP2119, UP2159, and UP2187) from which total RNA was extracted using the QIAamp RNA Blood Mini Kit (Qiagen, Hilden, Germany) and sequenced to generate an average of 108 million paired-end reads per sample. Quality control was performed with raw RNA-seq FASTQ files using FastQC[93], with summary reports generated by aggregating individual reports with MultiQC[94]. To remove ribosomal RNA (rRNA) contamination, we filtered the reads using SortMeRNA[95] with SILVA and RFAM databases[96,97]. A human reference genome index was generated using GRCh38 primary assembly GENCODE v47 (https://www.gencodegenes.org/human/). Reads were aligned, and gene-level counts were quantified using the STAR aligner[98], resulting in an average genome coverage of 9x. Gene count matrices were imported into RStudio for downstream analysis. Genes with low expression were filtered out using DESeq2, retaining only those with more than 10 counts in at least three samples[99]. To mitigate potential globin mRNA contamination reads aligning to globin genes were excluded from further downstream analyses[100,101]. Count normalisation was performed using the median of ratios method in DESeq2[99]. Gene-specific expressions were plotted for PP-SV patients against the non-PP-SV controls.

### Biallelic loss and somatic second hit identification in PP-SV presenting patients

LOH was inferred by TitanCNA snake workflow (TITAN) v1.17.1[18,50]. In brief, tumour purity and ploidy corrected copy number status was inferred from matched tumour WGS data of PP-SV presenting patients by TITAN. The gene region with the adjusted discrete copy number of one allele as zero was considered to have LOH status. Depending on the copy number of another allele (1,2 or ≥2), TITAN predicted LOH status to hemizygous LOH, copy neutral LOH and amplification LOH, respectively (Supplementary Data 3).

The number of somatic oncogenic driver variants in all patients were obtained from our previous study[18], accounted for coding or noncoding driver mutation, significantly recurrent breakpoint and gene-level copy number amplification or deletion. In addition, all oncogenic driver variants were assessed for their presence in PP-SV presenting patients, as second somatic hit to PP-SV disrupted genes.

### Inclusion and Ethics Statement

Local researchers are included in this research as co-directors for the Southern African Prostate Cancer Study (SAPCS) from study design to interpretation, including data ownership via leadership on the SAPCS Data Access Committee (DAC), which also includes meeting all criteria for full authorship. The SAPCS Directorship includes clinical (M.S.R.B., University of Pretoria, South Africa), urological (S.B.A.M., Sefako Magatho Health Sciences University, South Africa) and scientific leaders (V.M.H., which includes affiliation at University of Pretoria, South Africa). Southern African data and material for RNA-sequencing data generation

was accessed via the SAPCS DAC and through a fully executed collaborative research agreement (CRA), which includes shared funding.

**Reporting summary**

Further information on research design is available in the Nature Portfolio Reporting Summary linked to this article.

## Data availability

Access to published whole genome sequence data (Jaratlerdsiri et al. [18]) was made available via Data Access Committee (DAC) approval as outlined under the European Genome-Phenome Archive (EGA) [https://ega-archive.org] project-specific access policies under over-arching study EGAS00001006425, which includes the Southern African Prostate Cancer Study (SAPCS) Dataset at EGAD00001009067 and Garvan/St Vincent's Prostate Cancer Database at EGAD00001009066. The mapped RNA-sequencing data in BAM format have been deposited under study EGAS50000000702 with accession number EGAD50000000982. Access to the RNA-sequencing data may be requested via the SAPCS DAC and will be made available to researchers with appropriate feasibility and corresponding ethics approvals to ensure the safeguarding of patient genomic information (contact V.M.H.). Restrictions include (i) No transfer to third parties allowed, (ii) acknowledgement of the SAPCS in publications/presentations, (iii) a report of the results of the research to be provided to DAC after completion (or when requested), (iv) researchers cannot utilise the data for commercial purposes, or any other purposes not approved by the DAC, and (v) approval will not be given that excludes other researchers from accessing data. Data currently being used for capacity building in under-resourced studies across Sub-Saharan Africa will be given priority and at times, may be granted time-limited exclusive rights for no more than a two-year period. SV and related annotation data supporting the findings of this study are available within the main text, Supplementary information and source data. Previously published SV sites and their population variant allele frequencies are available in the dbVar database [https://www.ncbi.nlm.nih.gov/dbvar] [91], gene regions are available in the ENSEMBL database [https://www.ensembl.org] [86], gene sets at MSigDB [47] [https://www.gsea-msigdb.org/gsea/index.jsp] and cancer driver gene sets at COSMIC CGC [48] [https://cancer.sanger.ac.uk/census]. Source data are provided in this paper.

## Code availability

Software and scripts for DNA sequence read data collection are available at GitHub (https://github.com/Sydney-Informatics-Hub/Bioinformatics). FastQC 0.11.7, MultiQC 1.25.1, Trimmomatic 0.38, SortMeRNA 4.3.3 and STAR aligner 2.7.1a for RNA sequence data collection. The scripts for sequence read alignment and quality control are available on GitHub (https://github.com/Sydney-Informatics-Hub/Bioinformatics). All computational code for SV call set comparison and integration are available at GitHub (https://github.com/tgong1/StructuralVariantUtil) [102].

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

## Acknowledgements

We are forever grateful to the patients and their families who have contributed to this study; making this research possible. We acknowl-edge the contributions of the many clinical staff across the SAPCS (South Africa) and St Vincent's Hospital Sydney (Australia), who, over many years, have recruited patients and provided samples to these critical bioresources. We are additionally grateful to Dr Md Mehedi Hasan of the Ancestry & Health Genomics Laboratory at the University of Sydney for the preparation of RNA-sequencing samples, the staff at the Ramaciotti Genomics Facility at the University of New South Wales Sydney for RNA-sequencing data generation, as well as additional authors who contributed to the initial published genome profiling pro-ject (Jaratlerdsiri et al.)[18]. Genomic sequencing was supported by the National Health and Medical Research Council (NHMRC) of Australia through a Project Grant (APP1165762 to V.M.H.) and Ideas Grants (APP2001098 to V.M.H. and M.S.R.B.; APP2010551 to V.M.H.), with control genomic sequencing and analysis supported by the U.S.A. Congressionally Directed Medical Research Programmes (CDMRP) Prostate Cancer Research Programme (PCRP) HEROIC Consortium Award (PC210168 and PC230673, HEROIC PCaPH Africa1K to V.M.H. and M.S.R.B., as well as co-leads Professors Gail Prins, University of Illinois at Chicago, U.S.A. and Mungai Peter Ngugi, University of Nairobi, Kenya). Further analytical support and RNA expression analyses was provided by the U.S.A. CDMRP PCRP Idea Development Award (PC200390, TARGET Africa to V.M.H.), a U.S.A. National Institute of Health (NIH) National Cancer Institute (NCI) Award (1R01CA285772-01 to V.M.H.) and the U.S.A. Prostate Cancer Foundation (PCF) Challenge Award (2023CHAL4150 to V.M.H.). T.G. was further supported by the Office of China Postdoctoral Council International Postdoctoral Exchange Fellowship Programme (Talent-Introduction Programme, YJ20210053) and V.M.H. by the Petre Foundation via the University of Sydney Foundation, Australia.

## Author contributions

Conception and design: T.G. and V.M.H.; Financial support: V.M.H.; Methodology: T.G. and J.J. (SV interrogation), K.U. (RNA-seq interroga-tion); Formal analysis: T.G., J.J., K.U., W.J. and V.M.H.; Data curation: J.J., K.U., K.G. and W.J.; Participant recruitment, clinical data and sample collection: S.B.A.M., P.D.S., and M.S.R.B.; Supervision: V.M.H.; Writing-review, figures and editing: T.G., K.U. and V.M.H. Critical technical review: J.W. All authors have read and approved the final manuscript.

## Competing interests

The authors declare the following competing interest, that V.M.H. is a Member of Active Surveillance Movember Committee. The authors declare no other competing interests.
