## [Peer Review File · Nature Communications]

Rare pathogenic structural variants show potential to enhance prostate cancer germline testing for African men

Corresponding Author: Professor Vanessa Hayes

Version 0:

Reviewer comments:

Reviewer #1

(Remarks to the Author)

Men with African ancestry are at greater risk of aggressive prostate cancer. This study complements the previous work by authors where they performed tumor whole-genome sequencing and provides germline whole-genome sequencing data for 113 African and 57 European patients. They then identified germline structural variations and identified rare “potential pathogenic SVs” based on predicted functional impact. My biggest concern is the lack of control population with African ancestry to answer the very important question that are these SVs truly enriched in the patients that show prostate cancer. The potential pathogenic SVs can have many false positive pathogenicity predictions. Allele-specific annotation is also crucial because alleles with rearrangements in regions associated with cancers can be found segregating in the healthy karyotype population. This can be explained by the possibility that the allele is silenced, gene expression might not change significantly, or more complex scenarios of cell-type-specific activation/inactivation of genomic regions may occur. The description of genes predicted to be potentially pathogenic as it relates to prostate cancer are very vague and no substantial link of germline allele and tumor features is shown, and it likely can not be shown with the small sample size of the study. For example, the authors highlight RB1. RB1 is associated with lineage plasticity in advanced castration-resistant prostate cancer but the authors do not comment if the presence of RB1 alterations in germline is related to higher risk of lineage plasticity in castration-resistant disease. Thus, overall, this study lacks rigor.

(Remarks on code availability)

Reviewer #2

(Remarks to the Author)

I co-reviewed this manuscript with one of the reviewers who provided the listed reports as part of the Nature Communications initiative to facilitate training in peer review and appropriate recognition for co-reviewers.

(Remarks on code availability)

Reviewer #3

(Remarks to the Author)

This is a thought-provoking study by the authors and impactful to the field of PCa disparities. Rare and common structural variants (SVs) are important contributors to the genetics of numerous human diseases including prostate cancer (PCa). The authors are the first to investigate rare structural variants in PCa patients with ancestry disparities. The authors performed whole genome sequencing (WGS), structural variant (SV) calling, population genotyping, gene annotation and functional impact of SVs, and pathogenicity prediction. Within their results, they report the identification of African-private (eight known and three novel) and European-private (two known and two novel) potentially pathogenic (PP) SVs. In addition, three ClinVar-defined pathogenic or PP-SVs in (SLC3A1, OCA2, and PIGN) and 12 predicted PP-SVs, including seven known SVs in (SLC7A2, DNAJC15, COL4A2, SLC2A5, WASF1, MLH1, and RB1), and five novel SVs in (BCL2L11, BARD1, FOXP1, CTNNA1, and AK8-DST), suggesting that inherited SVs may constitute an under-appreciated contribution to PCa pathogenicity, especially in AA men with PCa. Thus, results support the conclusion that identifying novel and rare SVs may

potentially improve the detection rate for PCa germline testing, and in turn raise limitations for men of African descent inclusion and associated clinical care. Lastly, the methodology and analysis were robust. Thus, does not prohibit publication.

Comments: The manuscript was well written and is truly an interesting article that will push the field of PCa disparities forward. However, here are some comments:

Line 52 Rucaparib should be capitalized.

Line 165 $P < 0.001$ Space the p, the less than sign, and the value. Lowercase the p in P value. Lastly, keep consistent throughout the document.

Line 205 $AF = 0.03$, Space AF, the equal sign, and the value. Keep consistent throughout the document.

Line 276 The first time mentioning a gene make sure it is fully described (for example: Line 276 SLC3A1 is mentioned, however, later in Line 285 it is mentioned and fully described).

Major Critiques

The authors could benefit from adding a limitation section in the manuscript. It might be best to place that section at the bottom of the discussion, right before the conclusion.

(Remarks on code availability)

Version 1:

Reviewer comments:

Reviewer #1

(Remarks to the Author)

The authors respond that “there is currently no available control population data for southern African men without a prostate cancer diagnosis”. Below is a short list of notable studies that have reported whole genome sequencing of African individuals. We urge the authors to check these studies for samples that correspond to the south African population and can be used as controls. This would greatly improve the rigor of the study.

Fan, S., Kelly, D.E., Beltrame, M.H. et al. African evolutionary history inferred from whole genome sequence data of 44 indigenous African populations. *Genome Biol* 20, 82 (2019). <https://doi.org/10.1186/s13059-019-1679-2>

· This study sequenced the genomes of 92 individuals from 44 indigenous African populations.

Somineni HK, et al. Whole-genome sequencing of African Americans implicates differential genetic architecture in inflammatory bowel disease. *Am J Hum Genet.* 2021 Mar 4;108(3):431-445. doi: 10.1016/j.ajhg.2021.02.001. Epub 2021 Feb 17. PMID: 33600772; PMCID: PMC8008495.

· It describes a large-scale whole-genome sequencing reporting 1,644 healthy control Americans with African ancestry (African Americans).

Mallick, S., Li, H., Lipson, M. et al. The Simons Genome Diversity Project: 300 genomes from 142 diverse populations. *Nature* 538, 201–206 (2016). <https://doi.org/10.1038/nature18964>

· The Simons Genome Diversity Project (SGDP) reports deep genome sequences of 300 individuals from 142 populations and it includes 49 African individuals

Halldorsson BV, et al. The sequences of 150,119 genomes in the UK Biobank. *Nature.* 2022 Jul;607(7920):732-740. doi: 10.1038/s41586-022-04965-x. Epub 2022 Jul 20. PMID: 35859178; PMCID: PMC9329122.

· UK Biobank with an African cohort of 9,633 individuals.

The 1000 Genomes Project Consortium. An integrated map of genetic variation from 1,092 human genomes. *Nature* 491, 56–65 (2012). <https://doi.org/10.1038/nature11632>

· Although at low coverage the 1000 Genomes Project, analyzed 26 populations including sub-Saharan African ancestry

(Remarks on code availability)

Reviewer #2

(Remarks to the Author)

(Remarks on code availability)

Reviewer #3

(Remarks to the Author)

Authors adequately responded to my minor and major critiques and comments. More specifically, the authors addressed issues with ensuring nomenclature was consistent throughout the manuscript. Lastly, the limitation section was further discussed and elaborate.

(Remarks on code availability)

Version 2:

Reviewer comments:

Reviewer #1

(Remarks to the Author)

The authors have addressed my concerns in the revised version of the manuscript.

(Remarks on code availability)

Reviewer #2

(Remarks to the Author)

(Remarks on code availability)

Response to reviewers

Gong et al.,

Rare pathogenic structural variants show potential to enhance prostate cancer germline testing for African men

Responses in Blue

REVIEWER COMMENTS

Reviewer #1 (Remarks to the Author): Expert in computational cancer genomics and structural variant analysis

Men with African ancestry are at greater risk of aggressive prostate cancer. This study complements the previous work by authors where they performed tumor whole-genome sequencing and provides germline whole-genome sequencing data for 113 African and 57 European patients. They then identified germline structural variations and identified rare “potential pathogenic SVs” based on predicted functional impact. My biggest concern is the lack of control population with African ancestry to answer the very important question that are these SVs truly enriched in the patients that show prostate cancer. The potential pathogenic SVs can have many false positive pathogenicity predictions. Allele-specific annotation is also crucial because alleles with rearrangements in regions associated with cancers can be found segregating in the healthy karyotype population. This can be explained by the possibility that the allele is silenced, gene expression might not change significantly, or more complex scenarios of cell-type-specific activation/inactivation of genomic regions may occur. The description of genes predicted to be potentially pathogenic as it relates to prostate cancer are very vague and no substantial link of germline allele and tumor features is shown, and it likely can not be shown with the small sample size of the study. For example, the authors highlight RB1. RB1 is associated with lineage plasticity in advanced castration-resistant prostate cancer but the authors do not comment if the presence of RB1 alterations in germline is related to higher risk of lineage plasticity in castration-resistant disease. Thus, overall, this study lacks rigor.

Response to:

“My biggest concern is the lack of control population with African ancestry to answer the very important question that are these SVs truly enriched in the patients that show prostate cancer.”

Addressing lack of control populations: There is not only a scarcity of whole genome African data, but importantly there is currently no available control population data for southern African men without a prostate cancer diagnosis. We hope this improves going forward and we are excited that Africa is finally getting attention, we have addressed this as a limitation in the **Discussion** with the sentence that reads: “Additional study limitations related to our southern African cohort include (i) a lack of population-matched healthy controls or regionally relevant population wide whole genome data,”

Comment on rare functional SV enrichment: Unlike isolated populations, in particular those outside of Africa, African and in this case southern African populations, have not experienced a significant genetic bottleneck and in turn genetic drift [see Zeggini, E. Using genetically

isolated populations to understand the genomic basis of disease. *Genome Med.* **6**, 83 (2014)]. Additionally, it is well established that southern African populations have the greatest within population genetic diversity and in turn are most impacted by positive or purifying selection. Therefore, enrichment of rare functional variants in southern Africans is less likely to occur. [see: Xue, Y., Mezzavilla, M., Haber, M. *et al.* Enrichment of low-frequency functional variants revealed by whole-genome sequencing of multiple isolated European populations. *Nat Commun* **8**, 15927 (2017). <https://doi.org/10.1038/ncomms15927>.]

Response to:

“Allele-specific annotation is also crucial because alleles with rearrangements in regions associated with cancers can be found segregating in the healthy karyotype population. This can be explained by the possibility that the allele is silenced, gene expression might not change significantly, or more complex scenarios of cell-type-specific activation/inactivation of genomic regions may occur.”

Pathogenic germline variants can cause phenotypic consequences in different somatic tissues, but the vast majority are either hypomorphic, requiring loss-of-heterozygosity for full phenotypic penetrance, or are highly tissue-specific. While not possible due to the short read span of our Illumina sequencing, we agree that having allele-specific phasing information for the germline SVs would be very useful to assess the inheritance pattern and phenotype. Future studies with a larger haplotype-phased southern African cohort would greatly facilitate such relevant questions. We note that the germline SV data was derived from whole blood rather than normal tissue and as such cell-type variability is unlikely to be at play in this leucocyte derived data source. We have provided further clarification in the opening paragraph of the **Results**, stating that germline data was derived “**from whole blood**”.

We have further confirmed that variant allele frequency (VAF) for all relevant SVs per patient, as determined using manual inspection (IGV), fall within that expected for inherited germline variants. We have added the VAFs to **Table 2** and included a new **Supplementary Table 8**, while the following sentence has been added to the second last paragraph of the **Results**, “We further concur that all SVs are likely germline as their variant allele frequencies (VAFs) are as expected for heterozygous inheritance (average of 41.8%; range 30% to 51%, **Supplementary Table 8**).”

An additional paragraph was added under **Methods** to reflect VAF predictions as below:

PP-SV allele frequency predictions

The VAF of each PP-SV was calculated as fraction of altered sequencing reads within each SV region. Three types of altered sequencing reads were considered, including the reduced or increased sequencing reads due to DEL or DUP, discordantly aligned read-pairs and split reads. For DEL and DUP, the altered read count was calculated as the average read depth difference between SV region and its ± 10 kb regions. Total read count of DEL was measured as the average read depth of ± 10 kbp region to SV region, and that for DUP was the average read depth of SV region. For INV and TRA, the altered reads, including discordantly aligned reads and split-reads, within ± 150 bp of each breakpoint were manually inspected using IGV. The total read count, was measured as average sequencing read depth in ± 150 bp region to INV or TRA breakpoints, calculated using samtools (v1.6) *depth* command. The SV VAF can be underestimated due to one read can show both discordant read-pair and split-reads, while only counted once. In addition, non-reference read-pairs is possible to be aligned properly in SV region.

Response to:

“The description of genes predicted to be potentially pathogenic as it relates to prostate cancer are very vague and no substantial link of germline allele and tumor features is shown, and it likely can not be shown with the small sample size of the study. For example, the authors highlight RB1. RB1 is associated with lineage plasticity in advanced castration-resistant prostate cancer but the authors do not comment if the presence of RB1 alterations in germline is related to higher risk of lineage plasticity in castration-resistant disease.”

To further address potential functionality/causality of PP-SVs, we assessed for loss of heterozygosity (LOH) and a second gene-associated somatic hit in patient-matched tumour data and includes new **Supplementary data 3** (LOH) and new **Supplementary Table 9** (second hit), and further summarised in **Table 2**.

Additional text in red under Results:

Correlating PP-SVs with clinical features and tumour features of causality

Furthermore, tumour-matched samples from PP-SV presenting patients were assessed for biallelic loss or a second somatic hit, both potential indicators of gene-relevant causality⁵⁰. We used TITAN to infer for copy number (CN) loss as a prediction of loss of heterozygosity (LOH-CNL) due to deletion of wild-type allele, and CN neutral or gain LOH (LOH-CNN and LOH-CNG, respectively) due to accompanied duplication of the mutant allele⁵¹. After correction for tumour purity and ploidy (**Supplementary Data 3**), we identified LOH in five PP-SV presenting patients, including two with ClinVar validated pathogenic or likely pathogenic SVs and three presenting with ISUP 4-5 disease (**Table 2**). LOH-CNL was observed in a single patient SMU083 resulting in biallelic loss of *PIGN*. LOH-CNG was observed in two patients, indicating wild-type allele loss for *SLC3A1* and *SLC7A2* in patients SMU094 and KAL0054, respectively. LOH-CNN, indicating wild-type allele loss with amplification of tumour suppressor losses, was observed in *BCL2L11* and *BARD1* for patients KAL0101 and N0073.

Overall and irrespective of ancestry, patients with germline PP-SVs (14 African, 4 European) showed less oncogenic driver variants than non-PP-SV presenting cases (99 African, 53 European), although not statistically significant (250, range: 101-437 vs 316, range: 105-772). In addition, we found same gene second hit by somatic CN alterations including CN loss impacting *SLC3A1*, *SLC7A2* and two cautionary fusion PP-SVs in an African patient each (all ISUP ≥ 3 at diagnosis) and in turn CN gain impacting *OCA2*, *FOXP1* and *WASF1* in each of three African patients with advanced ISUP 5 disease (**Supplementary Table 9** and **Table 2**). Notably, the single European patient presenting with the cautionary PP-SV impacting *LTBP1*, with second hit somatic CN gain, underwent surgery at age 54 years for ISUP 5 disease.

Additional text in red under Discussion:

SCL3A1 over-expression has been associated with enhanced tumourigenesis in breast cancer, while blocking *SCL3A1* has suggestive therapeutic potential³⁹. Taken together, it is notable that somatic LOH-CNG, with second hit CN loss, was observed for the younger of the two *SLC3A1* PP-SV presenting African patients (SMU094).

OCA2 is a pigmentation gene with inherited mutations associated with oculocutaneous albinism⁵⁴. Polymorphisms have been associated with skin cancers⁵⁵, as well as clinical response and survival in breast cancer patients having received neoadjuvant chemotherapy⁵⁶.

Notably, the ISUP GG5 presenting pLoF germline SV African patient also with a second hit somatic *OCA2* CN gain.

Inherited *PIGN* mutations have been associated with multiple congenital anomalies-hypotonia-seizures syndrome and Fryns syndrome, with some mutations related to milder forms of clinical presentation^{57, 58}. *PIGN* is involved in the biosynthesis of glycosylphosphatidylinositol, which has been shown to suppress cancer chromosomal instability³⁷ through PIGN complexed spindle assembly checkpoint regulation³⁸, a common phenomenon in solid tumours⁵⁹. Biallelic *PIGN* loss is tentatively predicted in the tumour of the older aged presenting African patient.

And all genes where LOH/second hit events were observed (see **Main Text**). Also, in the final paragraph of the **Discussion**, before the conclusion, we make this observation (as a limitation). While LOH or second hits in the developing tumours added further possible causality to several candidate genes, one cannot ignore that LOH can occur by chance. Conversely, we cannot exclude for hypermethylation inactivation as a second hit for the remaining gene candidates.

Additional text in red under **Methods**:

Biallelic loss and somatic second hit identification in PP-SV presenting patients

LOH was inferred by TitanCNA snake workflow (TITAN) v1.17.1⁵¹, as previously described¹⁸. In brief, tumour purity and ploidy corrected copy number status was inferred from matched tumour WGS data of PP-SV presenting patients by TITAN. The gene region with the adjusted discrete copy number of one allele as zero was considered to have LOH status. Depending on the copy number of another allele (1,2 or ≥ 2), TITAN predicted LOH status to hemizygous LOH, copy neutral LOH and amplification LOH, respectively (**Supplementary Data 3**).

The number of somatic oncogenic driver variants in all patients were obtained from our previous study¹⁸, accounted for coding or noncoding driver mutation, significantly recurrent breakpoint and gene-level copy number amplification or deletion. In addition, all oncogenic driver variants were assessed for their presence in PP-SV presenting patients, as second somatic hit to PP-SV disrupted genes.

Reviewer #2 (Remarks to the Author): Early-Career Researcher co-reviewer

I co-reviewed this manuscript with one of the reviewers who provided the listed reports as part of the Nature Communications initiative to facilitate training in peer review and appropriate recognition for co-reviewers.

Response: Appreciated.

Reviewer #3 (Remarks to the Author): Clinical expert in prostate cancer genetics and ancestry-related health disparities

This is a thought-provoking study by the authors and impactful to the field of PCa disparities. Rare and common structural variants (SVs) are important contributors to the genetics of numerous human diseases including prostate cancer (PCa). The authors are the first to investigate rare structural variants in PCa patients with ancestry disparities. The authors performed whole genome sequencing (WGS), structural variant (SV) calling, population genotyping, gene annotation and functional impact of SVs, and pathogenicity prediction.

Within their results, they report the identification of African-private (eight known and three novel) and European-private (two known and two novel) potentially pathogenic (PP) SVs. In addition, three ClinVar-defined pathogenic or PP-SVs in (SLC3A1, OCA2, and PIGN) and 12 predicted PP-SVs, including seven known SVs in (SLC7A2, DNAJC15, COL4A2, SLC2A5, WASF1, MLH1, and RB1), and five novel SVs in (BCL2L11, BARD1, FOXP1, CTNNA1, and AK8-DST), suggesting that inherited SVs may constitute an under-appreciated contribution to PCa pathogenicity, especially in AA men with PCa. Thus, results support the conclusion that identifying novel and rare SVs may potentially improve the detection rate for PCa germline testing, and in turn raise limitations for men of African descent inclusion and associated clinical care. Lastly, the methodology and analysis were robust. Thus, does not prohibit publication.

Comments: The manuscript was well written and is truly an interesting article that will push the field of PCa disparities forward.

Response: We appreciate the recognition.

However, here are some comments:

Line 52 Rucaparib should be capitalized.

Response: Updated in text.

Line 165 $P < 0.001$ Space the p, the less than sign, and the value. Lowercase the p in P value. Lastly, keep consistent throughout the document.

Response: Updated in text and kept consistent through the text as “p-value”, shown in red.

Line 205 $AF = 0.03$, Space AF, the equal sign, and the value. Keep consistent throughout the document.

Response: Updated in text and kept consistent through the text, shown in red.

Line 276 The first time mentioning a gene make sure it is fully described (for example: Line 276 SLC3A1 is mentioned, however, later in Line 285 it is mentioned and fully described).

Response: Updated in text in red.

Major Critiques

The authors could benefit from adding a limitation section in the manuscript. It might be best to place that section at the bottom of the discussion, right before the conclusion.

Response: While the paragraph preceding the **Conclusion** addresses the study limitations (in black), here we expand on these limitations (in red) to not only include technical limitations, but also limitations associated with the study population.

Using short-read sequencing data for SV calling and genotyping remains a potential limitation, appreciating that SVs in difficult-to-sequence regions may have been overlooked⁷⁷. To ensure the highest possible accuracy of SV detection and population allele frequency estimation, we required high-confidence calls from two SV callers and high-quality genotype calls at both the

population- and individual-level, while all PP-SVs were visually inspected. Due to lack of available expression data, we were unable to validate the direct impact of identified PP-SVs and cautionary PP-SVs. While LOH or second hits in the developing tumours added further possible causality to several candidate genes, one cannot ignore that LOH can occur by chance. Conversely, we cannot exclude for hypermethylation inactivation as a second hit for the remaining gene candidates. Additional study limitations related to our southern African cohort include (i) a lack of population-matched healthy controls or regionally relevant population wide whole genome data, (ii) the on average older age at PCa presentation^{23, 78} and lack of PCa knowledge as related to family history⁷⁹, both criteria traditionally used for genetic testing, and (iii) a lack of African-relevant data in currently pathogenicity prediction databases. As a consequence of these limitations, our study calls for further African-inclusive efforts and for the establishment of guidelines for pathogenic SV identification using both short and/or long read sequencing approaches, making these methods accessible for routine multi-ethnic germline testing.

Rare pathogenic structural variants show potential to enhance prostate cancer germline testing for African men

Gong et al.,

RESPONSE TO REVIEWER COMMENTS R2

Response in **blue** with changes in main text of the manuscript in **red**.

Reviewer #1 (Remarks to the Author):

Comment: The authors respond that “there is currently no available control population data for southern African men without a prostate cancer diagnosis”. Below is a short list of notable studies that have reported whole genome sequencing of African individuals. We urge the authors to check these studies for samples that correspond to the south African population and can be used as controls. This would greatly improve the rigor of the study.

Fan, S., Kelly, D.E., Beltrame, M.H. et al. African evolutionary history inferred from whole genome sequence data of 44 indigenous African populations. *Genome Biol* 20, 82 (2019). <https://doi.org/10.1186/s13059-019-1679-2>

· This study sequenced the genomes of 92 individuals from 44 indigenous African populations.

Response: None of the 44 ethno-linguistic groups in this afore mentioned study represent Southern Bantu population identifiers and in turn Southern Bantu genetics.

Somineni HK, et al. Whole-genome sequencing of African Americans implicates differential genetic architecture in inflammatory bowel disease. *Am J Hum Genet.* 2021 Mar 4;108(3):431-445. doi: 10.1016/j.ajhg.2021.02.001. Epub 2021 Feb 17. PMID: 33600772; PMCID: PMC8008495.

· It describes a large-scale whole-genome sequencing reporting 1,644 healthy control Americans with African ancestry (African Americans).

Response: We have previously demonstrated that our Southern African men, representing Southern Bantu ethno-linguistic identifiers, show differing within Africa population substructure from African Americans with largely west African genetic fractions, with our most recent *Nature Commun.* publication.

Reference: Soh PXY, Mmekwa N, Petersen DC, Gheybi K, van Zyl S, Jiang J, Patrick SM, Campbell R, Jaratlerdseri W, Mutambirwa SBA, Bornman MSR, Hayes VM. Prostate cancer genetic risk and associated aggressive disease in men of African ancestry. *Nat Commun.* 2023 Dec 5;14(1):8037.

Here we not only report on the lack of genomic data across our study region, providing reference to the latest version of gnomAD (Figure 1A, red circles across the map of the world), but using structure plot analysis we demonstrate that southern African Bantu (which includes patients from this study as part of the SAPCS) have a different genetic substructure compared to west, east and central Africans and including African Americans (ASW), with predominantly west African genetic substructure (Figure 1B). Additionally, we have a paper under review that shows differing genetic substructure between south and southwest Bantu groups, while the limited number of southern African genomes available are largely restricted to non-Bantu

genomes representing the KhoeSan population (or San) identifier, representing a different genetic substructure.

Fig. 1 | Summary of genome-wide association studies (GWAS) in Africa and ancestral fractions for our dataset. **A** Map showing populations across Africa where GWAS studies have been conducted for prostate cancer, and sample locations and sizes of African ancestry from the Human Genome Diversity Project (HGDP) and 1000 Genomes Project (IKGP) subset of gnomAD v3.12²⁷ (red circles). Mortality rates of prostate cancer from GLOBOCAN 2020 are shown in the bottom

left bar plot, with the population size of men in the region indicated in brackets below the region name²⁸. Study references: ^aCook et al.¹⁰, ^bHarlemon et al.¹³, ^cTindall et al.¹², ^dDu et al.¹¹. **B** Admixture plot ($K = 5$, cross-validation error = 0.162) which was replicated in 10 out of 10 runs, including 1003 Africans, 20 Europeans, 20 Chinese individuals from the HGDP and IKGP subset of gnomAD with our dataset of 781 South Africans.

Mallick, S., Li, H., Lipson, M. et al. The Simons Genome Diversity Project: 300 genomes from 142 diverse populations. *Nature* 538, 201–206 (2016). <https://doi.org/10.1038/nature18964> · The Simons Genome Diversity Project (SGDP) reports deep genome sequences of 300 individuals from 142 populations and it includes 49 African individuals.

Response: Again, with the SGDP, of the 49 African genomes, only two are from a Southern Bantu ethnic group defined in this study as Tswana and belonging to the broader Sotho-Tswana ethno-linguistic group. Here our 113 Southern Bantu represent all four major linguistic groupings defined as Nguni, Sotho-Tswana, Venda and Tsonga. In the publication below, we have shown genetic substructure within the broad Southern Bantu identifier, but also ethno-linguistically driven variance in prostate cancer risk and aggressive disease presentation, with specific elevation for men from the Tsonga identifier.

Reference: Gheybi K, Mmekwa N, Lebelo MT, Patrick SM, Campbell R, Nenzhelele M, Soh PXY, Obida M, Loda M, Shirindi J, Butler EN, Mutambirwa SBA, Bornman MSR, Hayes VM. Linking African ancestral substructure to prostate cancer health disparities. *Sci Rep.* 2023 Nov 27;13(1):20909.

Figure 1

ADMIXTURE plot analysis for a representation of 780 Black South African men from the Southern African Prostate Cancer (SAPCS) for $K=4$ (replicated in 3 of 10 runs, cross-validation error of 0.369), demonstrating a unique predominant genetic fraction distinguishing the African ancestral ethno-linguistic groups defined as Venda (yellow), Tsonga (light blue), Nguni (dark blue) and Sotho-Tswana (red).

Halldorsson BV, et al. The sequences of 150,119 genomes in the UK Biobank. *Nature.* 2022 Jul;607(7920):732-740. doi: 10.1038/s41586-022-04965-x. Epub 2022 Jul 20. PMID: 35859178; PMCID: PMC9329122.
· UK Biobank with an African cohort of 9,633 individuals.

Response: UK BioBank assigned 9,633 individuals as African (XAF cohort), which were further identified as Yoruba in Ibadan, Nigeria based on microarray genotype data, shown in Figure below (Supplementary Figure 8, Halldorsson et al., 2022).

Supplementary Fig. 18 Cohort ADMIXTURE summaries.

Boxplots of ADMIXTURE-assigned ancestry per cohort. Horizontal lines within boxes represent medians, and tops and bottoms of boxes indicate 75th and 25th percentile values respectively. Whiskers (colored vertical lines) extend to minimum and maximum values. Overlaid black horizontal lines represent means, and black vertical lines extending above and below the horizontal lines represent +3 and -3 standard errors respectively. CEU (Northern Europeans from Utah), CHB (Han Chinese in Beijing), ITU (Indian Telugu in the UK), PEL (Peruvians in Lima), and YRI (Yoruba in Ibadan, Nigeria).

The 1000 Genomes Project Consortium. An integrated map of genetic variation from 1,092 human genomes. *Nature* 491, 56–65 (2012). <https://doi.org/10.1038/nature11632>
 · Although at low coverage the 1000 Genomes Project, analyzed 26 populations including sub-Saharan African ancestry

Response: The African ancestral 1000G-project genomes are represented in the genomAD database (see reference to Soh et al., *Nature Commun.* 2023, Figure 1).

Overall response: However, to address this important point and the lack of data, our team has recently generated deep WGS data (unpublished) for 49 regionally and population matched Southern Bantu disease-free and younger aged (<45 Years) control individuals (41 Female, age range: 18 - 44, 8 male, age range: 18 - 39). Additional text has been included in the **Results** in the last paragraph of the section headed “**Characterising ClinVar verified candidate potentially pathogenic SVs**” and reads as follows: “.....However, the *OCA2* PP-SV was observed at a low-frequency ($1\% < \text{MAF} \leq 5\%$) in our Southern African population-matched control data including whole genomes derived from 49 younger aged (< 45 years) largely female (41, age range: 18 - 44) over male participants (8, age range: 18 - 39). Further gene interrogation showed an additional pLoF TRA on *OCA2*, resulting in *DNAH9–OCA2* gene fusion (**Supplementary Table 3**).” Additional text at the end of the first paragraph under “**Correlating PP-SVs with clinical features and tumour features of causality**” reads as follows; “....Due to lack of available Southern African population-matched non-cancerous data, here we further interrogated 49 younger-aged non-cancerous Southern Africans for potentially pathogenic SVs within the 19 PP-SV defined candidate genes. Besides the previously discussed *OCA2* ClinVar defined “likely pathogenic” SV, no PP-SVs were identified within this cohort (**Table 1**). In turn, additional non-matched pLoF SVs were identified impacting *OCA2*, *BARD1* and *SLC2A2* (**Supplementary Table 3**).

This additional control PP-SV screening included a column added to **Table 1** and the inclusion of a new **Supplementary Table 3** (see below).

Supplementary Table 3. Gene-disruptive SVs defined as PP-SV or impacting the same gene as PP-SV in PCa patients.

Genes	Gene impact type ¹	chrom1	pos1	chrom2	pos2	SV type	MAF in control population ²
OCA2	pLoF	chr15	28,017,719	chr15	28,020,677	DEL	0.02
DNAH9 – OCA2	pLoF	chr17	11,694,612	chr15	27,913,931	TRA	0.29
BARD1	pLoF	chr2	214,726,966	chr2	214,727,028	DEL	0.12
BTBD7 - SLC2A5	pLoF	chr14	93,246,136	chr1	9,061,394	TRA	0.17

¹ Gene impact type based on gene annotation. pLoF: Potential loss-of-function. CG: Copy gain. IED: Intragenic Exon Duplication.

² Minor Allele Frequency (MAF) of SVs were estimated based on high-quality genotype calls only (detail in Methods).

An additional paragraph was also added to the **Methods** section under “**PP-SV allele frequency predictions**” which reads; “A control population, including 49 Southern Bantu matched disease-free individuals having undergone whole genome sequencing (Hayes Lab unpublished data) was made available for further PP-SV candidate gene interrogation. SV calling, genotyping and high-confidence SV filtering were implemented exactly same as PCa patients.”

Reviewer #2 (Remarks to the Author):

Comment: I co-reviewed this manuscript with one of the reviewers who provided the listed reports. This is part of the Nature Communications initiative to facilitate training in peer review and to provide appropriate recognition for Early Career Researchers who co-review manuscripts.

Response: We appreciate the ECR's contribution to this important initiative and wish them all the best as they develop their career independence.

Reviewer #3 (Remarks to the Author):

Comment: Authors adequately responded to my minor and major critiques and comments. More specifically, the authors addressed issues with ensuring nomenclature was consistent throughout the manuscript. Lastly, the limitation section was further discussed and elaborate.

Response: We appreciate the acknowledgement of our compliance and the reviewer's contribution to improving the quality of reporting.